# Finite-momentum Cooper pairing in proximitized altermagnets

Song-Bo Zhang [1,2,3] ✉, Lun-Hui Hu [4,5,6] ✉ & Titus Neupert[3]

Finite-momentum Cooper pairing is an unconventional form of super-conductivity that is widely believed to require finite magnetization. Alter-magnetism is an emerging magnetic phase with highly anisotropic spin-splitting of specific symmetries, but zero net magnetization. Here, we study Cooper pairing in metallic altermagnets connected to conventional $s$-wave superconductors. Remarkably, we find that the Cooper pairs induced in the altermagnets acquire a finite center-of-mass momentum, despite the zero net magnetization in the system. This anomalous Cooper-pair momentum strongly depends on the propagation direction and exhibits unusual sym-metric patterns. Furthermore, it yields several unique features: (i) highly orientation-dependent oscillations in the order parameter, (ii) controllable $0$-$\pi$ transitions in the Josephson supercurrent, (iii) large-oblique-angle Cooper-pair transfer trajectories in junctions parallel with the direction where spin splitting vanishes, and (iv) distinct Fraunhofer patterns in junctions oriented along different directions. Finally, we discuss the implementation of our predictions in candidate materials such as $RuO_2$ and $KRu_4O_8$.

Cooper pairs are fundamental to the phenomenon of super-conductivity and play a vital role in the emergence of its unique properties such as perfect electrical conductivity and Meissner effect[1]. Conventionally, Cooper pairs consist of electrons with opposite spins and momenta, thus carrying zero total momentum. The interplay of magnetism and superconductivity gives rise to various intriguing and exotic phenomena, making it one of the current research focuses in condensed matter physics[2–8]. In particular, a finite magnetization can induce Cooper pairs with finite center-of-mass momentum[9,10], which can be observed, e.g., in a 2D superconductor subjected to an applied magnetic field[11–14] or a ferromagnetic medium close to a superconductor[15–20] (see Fig. 1a, c for an illustration). The finite-momentum pairing manifests as an oscillating order parameter in real space. In Josephson junctions, the ground state usually has no phase difference across the junction and is referred to a 0-junction. However, the finite magnetization may produce an intrinsic $\pi$ phase difference,

forming a so-called $\pi$-junction[21,22]. Notably, a switchable $\pi$ state of Josephson junction holds important applications in superconducting circuits and qubits[23–25].

While it is widely believed that finite-momentum Cooper pairing requires a non-zero net magnetization, in this work, we challenge this by revealing that magnetic systems with zero net magnetization can after all support Cooper pairs with finite momentum by sacrificing uni-formity. As a proof-of-concept, we take altermagnetic metals as an example. Altermagnetism is an emerging magnetic phase beyond con-ventional ferromagnetism and antiferromagnetism and features highly anisotropic spin splitting in electronic bands with specific symmetry but zero net magnetization (e.g., a $d$-wave-like magnetism)[26–32] (see Fig. 1b). This novel phase may be caused by Fermi-surface instabilities[26–28]. It can also arise directly from symmetries of the crystal potential and does not require strongly correlated systems[29,31,32]. It breaks the combined sym-metry of translation and $C_2$ spin rotation that flips the spin (which is

[1]Hefei National Laboratory, Hefei, Anhui 230088, China. [2]International Center for Quantum Design of Functional Materials (ICQD), University of Science and Technology of China, Hefei, Anhui 230026, China. [3]Department of Physics, University of Zürich, Winterthurerstrasse 190, 8057 Zürich, Switzerland. [4]Department of Applied Physics, Aalto University School of Science, FI-00076 Aalto, Finland. [5]Center for Correlated Matter and School of Physics, Zhejiang University, Hangzhou 310058, China. [6]Department of Physics and Astronomy, The University of Tennessee, Knoxville, TN 37996, USA. ✉e-mail: songbozhang@ustc.edu.cn; hu.lunhui.zju@gmail.com

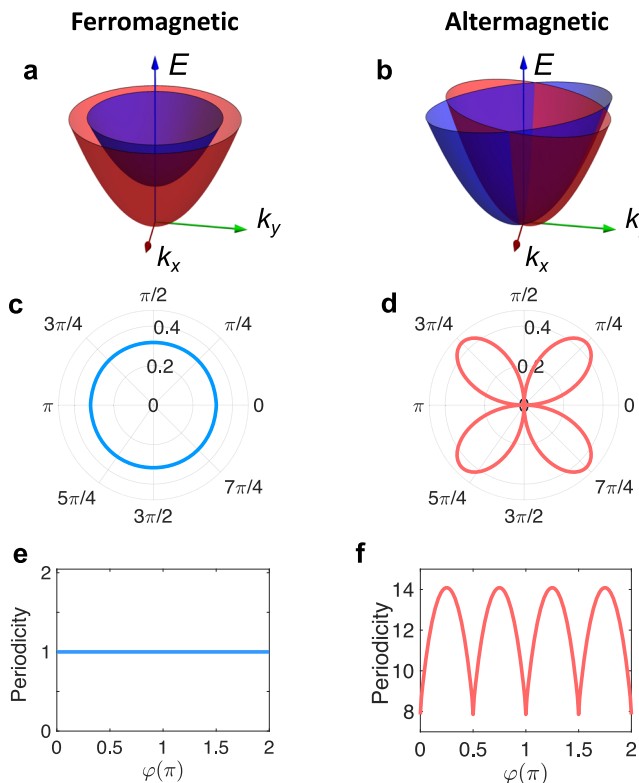

**Fig. 1 | Contrast of finite-momentum pairing in the ferromagnetic and altermagnetic metals. a, b** Schematics of the band structures of the ferromagnet and the altermagnet, respectively. Blue and red distinguish the two bands of opposite spins. In the calculation for the ferromagnet, we replace $Jk_xk_ys_z$ with a constant magnetization $Js_z$ in Eq. (1) and consider the long-wavelength limit with spatial rotation symmetry in the normal kinetic energy part. **c** Polarplots of the Cooper-pair momentum $q$ as a function of the propagation direction $\theta$ in the ferromagnet. **d** the same as (**c**) but for the altermagnet. The Cooper-pair momentum is strongly anisotropic and vanishes when propagating along a crystalline axis ($\theta = n\pi/2$ with $n \in \{0, 1, 2, 3\}$). **e** Periodicity (in units of $\pi\hbar/E_{ex}$) of the order parameter with respect to $y'/v_F$ as a function of junction orientation $\varphi$ in the ferromagnet. $y'$ is the distance from the ferromagnet-superconductor interface, $v_F$ is the Fermi velocity and $E_{ex}$ is the exchange energy of the ferromagnet. **f** Periodicity with respect to $\sqrt{\mu}y'$ as a function of $\varphi$ in the altermagnetic junction. Other parameters are $t = 1$ and $J = 0.8$.

required for classical collinear antiferromagnets), but preserves a joint symmetry of spatial rotation and $C_2$ spin rotation. Notably, altermagnetism has been found to exhibit many intriguing properties and functionalities, such as high-efficiency spin current generation[33], giant tunneling magnetoresistance[34,35] and anomalous Hall effect[36], thereby arousing considerable theoretical and experimental interest. Importantly, it has also been discovered in a growing number of collinear magnetic materials[28–32,36–41] including $RuO_2$[28,31], $KRu_4O_8$[31], $\kappa$-Cl[29,37] and $Mn_5Si_3$[42].

In this work, we study systematically the Cooper pair propagators in an altermagnetic metal and the superconducting proximity effect in planar junctions formed by the altermagnet and conventional $s$-wave superconductors. Remarkably, we find that the proximity-induced Cooper pairs in the altermagnet receive a finite momentum although the system has zero net magnetization. Such anomalous Cooper-pair momentum strongly depends on the propagation direction of the Cooper pair. It exhibits a highly anisotropic symmetric pattern and vanishes in particular directions (see Fig. 1d), as inherited from the intrinsic anisotropic spin splitting of the altermagnet. Moreover, it manifests several unique and measurable features: first, it gives rise to damped periodic oscillations in the order parameter as a function of doping in the altermagnet or

distance from the altermagnet-superconductor (AM-SC) interface, which occurs for any junction orientation. The periodicity and decaying behavior depend strongly on the junction orientation, contrasting with those in ferromagnetic junctions (see Fig. 1e, f).

Second, the finite-momentum pairing results in anomalous 0-$\pi$ transitions in a planar Josephson junction by modulating the doping in the altermagnet, the length or orientation of the junction, which occurs in the absence of a net magnetization. Note that a $\pi$-junction may also occur in Josephson junctions formed by antiferromagnets, which however requires odd layers of the antiferromagnet and thus a finite magnetization[43]. Third, we find that when the junction is along the direction where the spin splitting of the altermagnet is maximized, the superconducting transport is dominated by Cooper pairs moving along the junction direction. Whereas when the junction is along the direction where the spin splitting vanishes, the transport is dominated by Cooper pairs moving at large oblique angles from the junction direction. As a result, when reducing junction width, the current density in the former junction remains qualitatively unchanged, whereas it changes dramatically in the latter junction. To our knowledge, this effect has not been seen in previously known ferromagnetic and antiferromagnetic counterparts. Finally, we demonstrate that the distinct dominant Cooper-pair transfer trajectories in junctions oriented along different directions also result in different Fraunhofer interference patterns when subjected to an external magnetic field.

## Results
### Effective model
In alternagnets, the spin splitting of electronic bands changes sign in momentum space and the net magnetization is zero due to the presence of rotational symmetry. To illustrate our main results, we consider an altermagnetic metal with $d$-wave-like magnetism in two dimensions (2D). This can be realized, for instance, in thin films of $RuO_2$ and $KRu_4O_8$[32]. In the long-wavelength limit, the altermagnet can be described by[35]

$$\mathcal{H}(\mathbf{k}) = t(k_x^2 + k_y^2) + Jk_xk_ys_z, \tag{1}$$

where $\mathbf{k} = (k_x, k_y)$ is the wavevector, the Pauli matrices $\{s_x, s_y, s_z\}$ act on spin space, $t$ parameterizes the usual kinetic energy and is taken to be the unit of energy, i.e., $t = 1$, $J$ is the strength of altermagnetic order arising from the anisotropic exchange interaction[31,32]. The spin component in the $z$-direction is a good quantum number. Note, however, that our main results of finite-momentum pairing, order parameter and Josephson supercurrent, discussed below, hold for the case with the spin polarization in other directions. Without loss of generality, we work in the eigenbasis of $s_z$. The model respects $[C_2||C_{4z}]$ symmetry, i.e., a four-fold rotation in real space ($(x, y) \rightarrow (y, -x)$) together with a two-fold rotation in spin space ($(\uparrow, \downarrow) \rightarrow (\downarrow, \uparrow)$), which is indicated by the relation $s_y\mathcal{H}^*(k_x, k_y)s_y = \mathcal{H}(k_y, -k_x)$. Thus, the system has zero net magnetization. Moreover, the two spin-polarized bands of the model are given by $\epsilon_\eta(\mathbf{k}) = t(k_x^2 + k_y^2) + \eta Jk_xk_y$, where $\eta = \pm 1$ distinguishes spin-up and spin-down, respectively. It has a band structure similar to those for $KRu_4O_8$ and $RuO_2$[31,33] (see Fig. 1b). The spin splitting ($J_zk_xk_y$) is highly anisotropic and vanishes along the $k_x$- and $k_y$-axes, leading to the $d$-wave-like magnetism.

We note that the direction of the altermagnetic order may deviate from the crystal axis by an angle $\alpha$, which can be described by rewriting the altermagnetic term as $J[\cos(2\alpha)k_xk_y + \sin(2\alpha)(k_x^2 - k_y^2)/2]s_z$ in Eq. (1). This deviation will only cause an angle shift of $2\alpha$ in the propagation direction dependence of the Cooper-pair momentum and hence in the junction orientation dependence of the order parameter and Josephson supercurrent. It does not alter our main results qualitatively. For concreteness, we consider $\alpha = 0$ and focus on the realistic case with $|J| \lesssim 1$ in the following.

## Finite Cooper-pair momentum

In a ferromagnetic metal with proximity-induced superconductivity, the Fermi surface splitting of opposite spins leads to a momentum of Cooper pairs, which is nonzero in any direction (Fig. 1c). We ignore spin-orbit coupling which is typically small in ferromagnetic systems. In contrast, because of the sign-changing nature of spin splitting and vanishingly small net magnetization, unique physics arises in the altermagnet with superconductivity, which we demonstrate numerically and analytically below.

We analyze the Cooper-pair propagator[44] to study finite-momentum pairing and supercurrents induced in the altermagnet. We consider $s$-wave spin-singlet pairing, which is the case most easily realized experimentally. The Cooper-pair propagator, i.e., the simultaneous propagation of two electrons with opposite spins from one position $\mathbf{r}_1$ to another position $\mathbf{r}_2$ at zero temperature, can be calculated as the Cooperon bubble diagram[45], yielding

$$D(\mathbf{r}_2;\mathbf{r}_1) = \frac{(J_+ J_-)^{3/2}}{\pi^2 r^2 (J_+ + J_-)}(e^{iqr} + e^{-iqr}). \tag{2}$$

where $J_\pm = 1/\sqrt{2 \pm J\sin(2\theta)}$, $\theta$ is the angle between the propagation direction $\hat{\mathbf{r}}$ and the $x$-axis, $q = \sqrt{2\mu}(J_+ - J_-)$, and $\mu$ is the chemical potential of the altermagnet and tunable by a gate voltage. We have denoted by $\mathbf{r} = \mathbf{r}_2 - \mathbf{r}_1$ the displacement of Cooper pair. We provide more details of derivation in the Methods and Supplementary Information[45].

Strikingly, we observe a finite momentum $q$ of the Cooper pair from the Cooper-pair propagator in Eq. (2). The Cooper-pair momentum comes in pairs with opposite values $(q, -q)$, due to the rotational symmetry of the system. Moreover, it exhibits a fourfold rotational symmetry in the propagation direction $\theta$, inherited from the altermagnet. Its magnitude is maximized when the propagation is diagonal to the crystalline axes, i.e., $\theta = \pi/4 + n\pi/2$ with $n \in \{0, 1, 2, 3\}$, whereas vanishes when the propagation is along a crystalline axis, i.e., $\theta = n\pi/2$ (see Fig. 1d). For small $|J| \ll 1$, $q$ is approximately

$$q \approx J\sqrt{\mu}\sin(2\theta) = 2J\sqrt{\mu}xy/r^2. \tag{3}$$

Note that the Cooper-pair momentum occurs in the absence of a net magnetization and is proportional to the square root of the chemical potential $\sqrt{\mu}$ in the altermagnet, in sharp contrast to that in ferromagnets which requires a finite magnetization and decreases with increasing $\mu$[2,15,16]. As a result of this anomalous anisotropic momentum, the Cooper pair exhibits damped oscillations as it moves in real space, which is sensitively dependent on the propagation direction. This anisotropic behavior further gives rise to the unique features in the proximity-induced order parameter, Josephson supercurrent, and transfer trajectories of Cooper pairs in junction systems, as we illustrate below.

## Proximity-induced order parameter

Equipped with the Cooper-pair propagator, we first study the order parameter induced in the altermagnet connected to an $s$-wave superconductor. To this end, we consider a planar AM-SC junction with the interface at $y' = 0$ along the $x'$-direction, as sketched in the inset of Fig. 2a. The coupling of the superconductor and altermagnet $\lambda$ is constant along the interface. Here, the superscript $'$ indicates that the interface coordinate $x'$ is rotated by an arbitrary angle $\varphi$ from the crystalline $x$-axis of the altermagnet. The resulting local order parameter can be calculated from the Cooper-pair propagator as

$$\langle|\Psi(\mathbf{r}')|\rangle = \lambda \int_{-W/2}^{W/2} dx_1' D(\mathbf{r}';x_1',0), \tag{4}$$

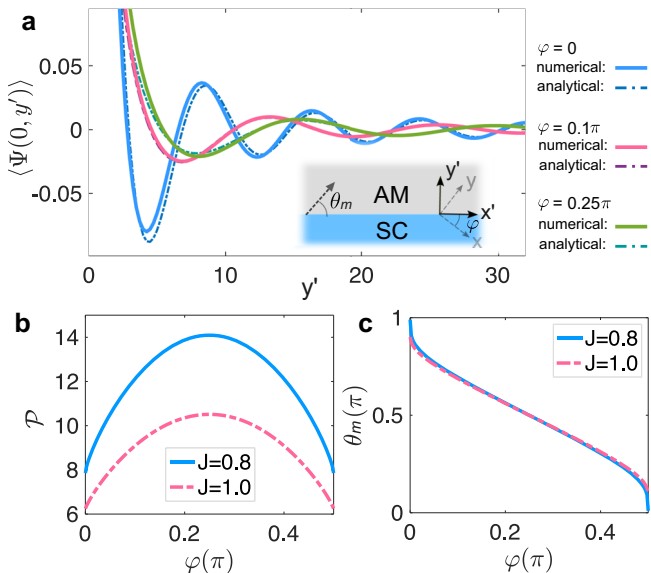

**Fig. 2 | Local order parameter near the altermagnet-superconductor (AM-SC) interface. a** $\langle|\Psi(0,y')|\rangle$ (in units of $\lambda/\pi^2$) as a function of distance $y'$ from the superconductor for junction orientations $\varphi = 0$, $0.1\pi$, and $0.25\pi$, respectively. We take $J = 0.8$ for illustration. The dashed curves are the plots of the formula in Eq. (5). Inset depicts the AM-SC junction with the interface at $y' = 0$ along $x'$-direction. **b** Periodicity $\mathcal{P}$ in $\langle|\Psi(0,y')|\rangle$ with respect to $\sqrt{\mu}y'$ as a function of $\varphi$. **c** Angle $\theta_m$ of dominant propagation trajectories [sketched in the inset of (**a**)] as a function of $\varphi$. Other parameters are $\mu = 1.0$ and $W = 1000$.

where $\mathbf{r}' = (x', y')$ and $W$ is the junction width. We have ignored the side boundary effect, which is justified for wide junctions $W \gg y'$. For large widths and chemical potentials, $W \gg y' \gg 1/(\sqrt{\mu}J)$, we find that $\langle|\Psi(\mathbf{r}')|\rangle$ becomes independent of $W$ and position $x'$ along the interface[45]. However, it exhibits damped oscillates around zero with distance $y'$ from the interface, as shown in Fig. 2a. These behaviors occur for any junction orientation. Under these considerations, we derive $\langle|\Psi(\mathbf{r}')|\rangle \approx \langle|\Psi(0, y')|\rangle$ as

$$\langle|\Psi(\mathbf{r}')|\rangle \approx \frac{\lambda \Upsilon_{\theta_m}}{(y')^{3/2}\mu^{1/4}}\cos\left(\mathcal{F}_{\theta_m}\sqrt{\mu}y' + \frac{\pi}{4}\right), \tag{5}$$

where $\Upsilon_{\theta'} = (2J_+'J_-'/\pi)^{3/2}/[(J_+' + J_-')|\partial^2\mathcal{F}_{\theta'}/\partial\theta'^2|^{1/2}]$, $J_\pm' = 1/\sqrt{2 \pm J\sin(2\theta' + 2\varphi)}$ and $\mathcal{F}_{\theta'} = \sqrt{2}\csc\theta'(J_-' - J_+')$. The angle $\theta_m$ is given by the minimum point of $\mathcal{F}_{\theta'}$. Physically, it is the angle of the Cooper-pair trajectory whose propagator dominates the integral in Eq. (4). Particularly, $\theta_m = \pi/2$ indicates that the dominant trajectory is normal to the AM-SC interface, while $\theta_m = 0$ (or $\pi$) indicates that the dominant trajectory is approximately parallel to the interface. We discuss this picture in more detail later. We provide more details of derivation in the Supplementary Information[45].

From Eq. (5), we see that the amplitude of $\langle|\Psi(\mathbf{r}')|\rangle$ decays from the AM-SC interface as $\sim(y')^{-3/2}$. This decay stems from the fact the Cooper-pair propagator decays with propagation distance $r'$ as $\sim(r')^{-2}$ [cf. Eq. (2)] and from the strong interference between the Cooper-pair propagators from the superconductor. The magnitude also decreases slowly as $\sim\mu^{-1/4}$ due to the interference between the propagators.

On top of the decay, $\langle|\Psi(\mathbf{r}')|\rangle$ oscillates periodically as $\sqrt{\mu}y'$ increases. The periodicity can be written as

$$\mathcal{P} = 2\pi/|\mathcal{F}_{\theta_m}|. \tag{6}$$

It decreases monotonically with increasing $|J|$. More interestingly, $\mathcal{P}$ increases monotonically as we rotate the junction orientation from

$\varphi = 0$ to $\pi/4$, as shown in Fig. 2b. At $\varphi = 0$, we find that the propagator dominating $\langle |\Psi(\mathbf{r'})| \rangle$ moves in a trajectory nearly parallel to the interface, i.e., $\theta_m \approx 0$. Hence, we have $\mathcal{F}_{\theta_m} = -J$. At $\varphi = \pi/4$, we find instead $\theta_m = \pi/2$, indicating that the dominant propagator moves in the trajectory normal to the AM-SC interface. Accordingly, $\mathcal{F}_{\theta_m} = \sqrt{2/(2+J)} - \sqrt{2/(2-J)} \approx -J/2$. Thus, we have $\mathcal{P}|_{\varphi = \pi/4}/\mathcal{P}|_{\varphi = 0} \approx 2$. $\langle |\Psi(\mathbf{r'})| \rangle$ oscillates more rapidly in junctions along a crystalline axis ($\varphi = 0$) than that diagonal to crystalline axes ($\varphi = \pi/4$). These behaviors are well corroborated by numerical calculations in Fig. 2a. It is also worth noting that $\langle |\Psi(\mathbf{r'})| \rangle$ is a periodic function of junction orientation $\varphi$ with a period $\pi/2$. For given large distance $y'$ and filling in the altermagnet, $\langle |\Psi(\mathbf{r'})| \rangle$ even oscillates around zero with $\varphi$ within a period. This angular dependence of the order parameter is a direct consequence of the anisotropic momentum of Cooper pairs in the altermagnet [cf. Eq. (3)] and is one of the central predictions in this work. It is generic and independent of the details of the superconductor and junction interface.

## Anomalous 0-π transitions in the supercurrent

The unique finite-momentum pairing in the altermagnet also manifests as interesting transport signatures in Josephson junctions. We consider a planar Josephson junction formed by sandwiching the altermagnet with length $L$ by two superconductors along the $y'$-direction. Similar to the AM-SC junction, we assume constant coupling amplitudes $\lambda_j$ (with $j \in \{1, 2\}$) along each interface, $\lambda_j(x') = \lambda_j e^{i\phi_j}$, where $\phi_j$ is the pairing phase in the $j$-th superconducting lead. The supercurrent can be evaluated as the derivative of free energy with respect to the pairing phase difference $\delta_\phi = \phi_1 - \phi_2$ across the junction. In terms of Cooper-pair propagators, the $\delta_\phi$-dependent part of free energy can be written as[45]

$$F_{\delta_\phi} = -4\lambda_1\lambda_2 \int dx_1' dx_2' \, \mathrm{Re}\left[e^{i\delta\phi} D(x_2', L; x_1', 0)\right]. \quad (7)$$

Plugging Eq. (2) into Eq. (7), we obtain the supercurrent as $I_s(\delta_\phi) = (e/\hbar)\partial F_{\delta_\phi}/\partial \delta_\phi = I_c \sin(\delta_\phi)$, where the critical current reads

$$I_c = \frac{4e}{\hbar} \lambda_1\lambda_2 \int_{-W/2}^{W/2} dx_1' dx_2' D(x_2', L; x_1', 0). \quad (8)$$

A positive $I_c$ corresponds to a 0-junction, whereas a negative $I_c$ indicates a $\pi$-junction where the system has an intrinsic $\pi$ phase difference across the junction at the ground state. Here, we ignore the correction from side boundary reflections. This is justified for $L \lesssim W$, as we show in the Supplementary Information[45]. For large widths and chemical potentials $W \gg L \gg 1/(\sqrt{\mu})$, we can evaluate $\int dx_1' D(x', L; x_1', 0)$ in a similar way as the order parameter. It turns out to be constant in $x'$. This allows us to find $I_c$ analytically as

$$I_c = \frac{4e}{\hbar} \frac{\lambda_1\lambda_2 W}{L^{3/2}\mu^{1/4}} \Upsilon_{\theta_m} \cos\left(\mathcal{F}_{\theta_m}\sqrt{\mu}L + \frac{\pi}{4}\right). \quad (9)$$

The agreement between the formula in Eq. (9) and numerical calculations is shown in Fig. 3. It is interesting to note that Eq. (9) takes the same form as the order parameter in the altermagnet, cf. Eq. (5). This suggests that the measurement of $I_c$, as a function of junction length, chemical potential, and junction orientation can give access to the information about the proximity-induced order parameter, such as its dependence on position (measured from the AM-SC interface), chemical potential and junction orientation.

Strikingly, $I_c$ exhibits pronounced oscillations around zero as a function of junction length $L$ and chemical potential $\mu$ in the altermagnet, as shown in Fig. 3a, c. The periodicity in $L$ is given by $\mathcal{P}_y = 2\pi/(\sqrt{\mu}\mathcal{F}_{\theta_m})$, while the periodicity in $\sqrt{\mu}$ reads $\mathcal{P}_\mu = 2\pi/(L\mathcal{F}_{\theta_m})$. They are the same as those in the order parameter. Thus, similarly, $I_c$ oscillates more rapidly in junctions along a crystalline axis where the

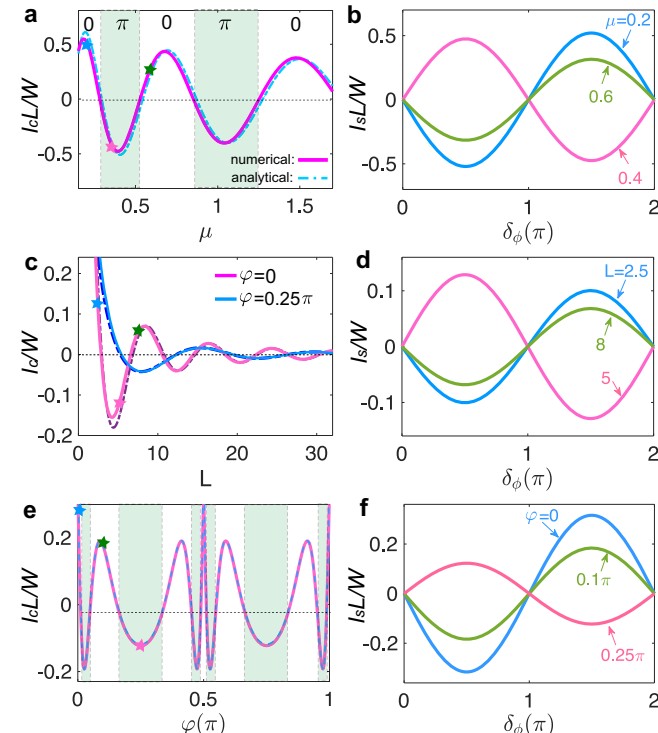

**Fig. 3 | Josephson 0-π transition. a** Critical supercurrent density $I_cL/W$ ($I_c$ is in units of $4e\lambda_1\lambda_2/\pi^2\hbar$) as a function of $\mu$ for $\varphi = 0$ and $L = 20$. The dashed curve is the plot of Eq. (9) with the same parameters. $I_c$ oscillates around 0 as $L$ increases. The light green shadows represent the $\pi$-junction regions as indicated. **b** Current-phase relation for $\mu = 0.2$, 0.4 and 0.6, respectively [marked by the colored stars in (**a**)]. **c** $I_c/W$ as a function of junction length $L$ for $\varphi = 0$ and $\pi/4$, respectively. We take $\mu = 1$ for illustration. The dashed curves are the plots of Eq. (9) for $\varphi = 0$ and $\pi/4$, respectively. **d** Current-phase relation for $L = 2.5$, 5, and 8, respectively [marked by the colored stars in (**c**)]. **e** $I_cL/W$ as a function of $\varphi$ for $\mu = 1$ and $L = 25$. The dashed curve is the plot of Eq. (9) with the same parameters. The light green shadows represent the $\pi$-junction regions. **f** Current-phase relation for $\varphi = 0$, $0.1\pi$ and $0.25\pi$, respectively [marked by the colored stars in (**e**)]. Other parameters are the same as Fig. 2 for all plots.

spin splitting of the altermagnet vanishes. In addition, $I_c$ is a periodic function of junction orientation $\varphi$ with period $\pi/2$. For large given $L$ and $\mu$, we also observe oscillations of $I_c$ around zero as a function of $\varphi$ within a period (see Fig. 3e). The oscillations become denser when $\varphi$ is close to $n\pi/2$ with $n$ being an integer because $\mathcal{F}_{\theta_m}$ changes faster there. Experimentally, one could fabricate curved devices with a series of superconducting lead pairs, similar to those used for anisotopic magnetoresistance measurements[46,47], which allows an effective rotation of the junction orientation.

As discussed earlier, positive (negative) $I_c$ corresponds to a 0 ($\pi$) state of the Josephson junction. These oscillations indicate 0-π transitions of the Josephson junction when we vary the junction length $L$, chemical potential $\mu$, or junction orientation $\varphi$, as shown in Fig. 3b, d, f. These results are also independent of the details of the $s$-wave superconductors and AM-SC interfaces (note that the interface couplings $\lambda_{1,2}$ only alter the magnitude of the current). It is also worth noting that the $\pi$-junction and 0-π transitions in the altermagnetic junction occur in the absence of a net magnetization. This is again in sharp contrast to ferromagnetic or antiferromagnetic junctions where a $\pi$-junction requires a finite net magnetization[15–20,43,48,49].

## Dominant Cooper-pair transfer trajectory

Our analytical result in Eq. (9) illustrates not only the anomalous 0-π transitions, but also intriguing features in the main trajectory direction of Cooper pairs transferring across the junction. The dominant Cooper-

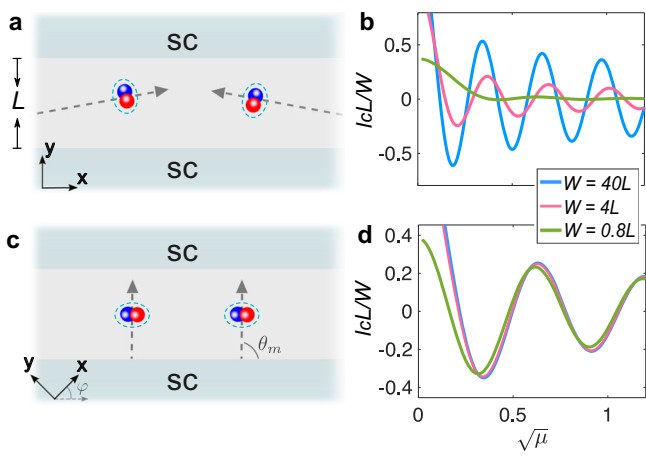

**Fig. 4 | Dominant transfer trajectory of Cooper pairs. a** Schematic of Cooper-pair transfer in junctions along *x*- or *y*-axis. The superconducting transport is dominated by Cooper pairs moving at large oblique angles (sketched by dashed arrows). **b** $I_c L/W$ as a function of $\sqrt{\mu}$ for $\varphi = 0$ and $W = 40L$, $4L$ and $4L/5$, respectively. **c** Schematic of Cooper-pair transfer in junctions along a diagonal (i.e. [11] or [1$\bar{1}$]) direction. The transport is dominated by Cooper pairs moving in the junction direction. **d** $I_c W/L$ as a function of $\sqrt{\mu}$ for $\varphi = \pi/4$ and $W = 40L$, $4L$ and $4L/5$, respectively. In (**b**, **d**), $L = 25$ and other parameters are the same as Fig. 2.

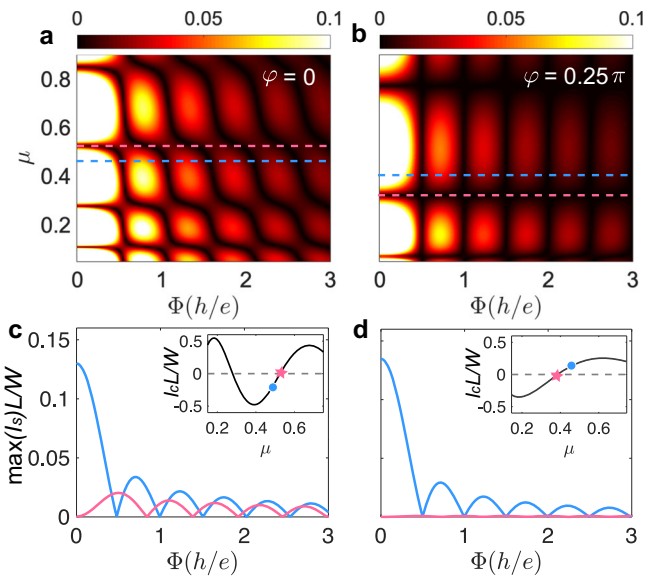

**Fig. 5 | Fraunhofer interference patterns. a** Maximal supercurrent $\max(I_s)$ ($I_s$ is in units of $4e\lambda_1\lambda_2/\pi^2\hbar$) as a function of chemical potential $\mu$ and external perpendicular magnetic flux (field) $\Phi = BWL$ (in units of magnetic flux quantum $h/e$) in a Josephson junction oriented along *x*- or *y*-axis. **b** The same as (**a**) but for a junction oriented with $\varphi = \pi/4$. **c** Fraunhofer patterns at a 0-$\pi$ transition point (pink) or away from the transition points (blue) in a junction oriented along *x*- or *y*-axis. The corresponding chemical potentials are marked by the colored markers in the inset. The two curves correspond to the two dashed line cuts in (**a**), respectively. **d** The same as (**c**) but for a junction oriented with $\varphi = \pi/4$. Other parameters are the same as Fig. 3.

pair transfer trajectory is described by the angle $\theta_m$, as we mentioned before. In Fig. 2c, we calculate $\theta_m$ as a function of junction orientation $\varphi$. We find that $\theta_m$ changes monotonically from 0 (or equivalently $\pi$) to $\pi/2$ as we rotate the junction from the direction where the splitting of the altermagnetic vanishes (i.e., crystalline axis, $\varphi = n\pi/2$ with $n \in \{0, 1, 2, 3\}$) to the diagonal direction where the spin splitting maximizes (i.e., $\varphi = \pi/4 + n\pi/2$). This indicates that the main transfer direction of Cooper pairs changes substantially. In particular, when the junction is oriented diagonal to the crystalline axes, we find $\theta_m = \pi/2 + n\pi$. Thus, the transport is dominated by Cooper pairs that move in the junction direction (see a sketch in Fig. 4c). In contrast, when the junction is parallel to a crystalline axis, we find $\theta_m = n\pi$. The transport is instead dominated by Cooper pairs that move at large oblique angles from the junction direction (see a sketch in Fig. 4a). We remark that such a large-oblique-angle Cooper-pair transfer is closely associated with the vanishing of Cooper-pair momentum in specific directions and is absent in ferromagnetic or antiferromagnetic junctions.

As a result of the distinct dominant transfer trajectories of Cooper pairs, the transport properties in junctions along and diagonal to the crystalline axes respond differently to the change of junction width $W$. Specifically, for junctions along a crystalline axis, the amplitude of the current density $I_c/W$ is significantly suppressed by reducing $W$ since fewer Cooper pairs move at large oblique angles. The oscillations in the order parameter and hence 0-$\pi$ transitions of the supercurrent are also strongly suppressed for small $W \lesssim L$. In contrast, for junctions diagonal to the crystalline axes, $I_c/W$ is nearly insensitive to the change of $W$. Accordingly, the oscillations and 0-$\pi$ transitions can be observed even for $W \lesssim L$. These results are well confirmed by our numerical calculations in Figs. 4b, d, where we perform the direct integration of $x'$ and $x_1'$ from $-W/2$ to $W/2$ in Eq. (8).

### Fraunhofer interference pattern

Finally, we apply a perpendicular magnetic field $B$ to the Josephson junction and study how it influences the supercurrent. The application of a magnetic field will induce spatial variations in the phase of pairing potentials along the AM-SC interfaces, thus significantly altering the interference between the Cooper-pair propagators and supercurrent across the junction. We provide the details of the calculation in the Supplementary Information[45].

In Fig. 5, we compute numerically the Fraunhofer interference patterns which measure the maximum supercurrent $\max(I_s)$ in response to the applied magnetic field, for varying chemical potential. It is striking to see that the junctions oriented along the maximum and vanishing Fermi surface spin-splitting directions exhibit distinct Fraunhofer patterns (see Fig. 5a, b). Explicitly, in the junction along the direction where the Fermi surfaces are most split ($\varphi = \pi/4$), we observe a conventional Fraunhofer pattern, in which the maximum supercurrent is located at zero field (see Fig. 5b, d). This feature is similar to that in ordinary Josephson junctions as the dominant Cooper-pair transfer trajectory consistently aligns with the junction direction. In contrast, at the 0-$\pi$ transition points of the junction along the direction where the spin-splitting vanishes (i.e., *x*- or *y*-axis), a finite supercurrent can be induced and enhanced by the applied magnetic field, as shown in Fig. 5c. This result is closely related to the fact the dominant Cooper-pair propagators are moving at large oblique angles across the junction. Note that in this case, the local critical-current density $j_c(x')$ may significantly vary with the change of the interface coordinate $x'$, which is different from ordinary planar junctions where Cooper pairs propagate mainly in the junction direction and thus $j_c(x')$ is approximately a constant. These contrast Fraunhofer patterns provide us with another compelling signature to detect the unique superconducting transport properties of the altermagnetic junctions.

## Discussion

To summarize, we have shown that Cooper pairs in the altermagnet acquire a finite momentum despite the system having zero net magnetization. This anomalous momentum is highly anisotropic with respect to the direction of Cooper-pair propagation. We have further shown that it gives rise to several unique features: (i) The order parameter oscillates with the gate voltage on the altermagnet and/or with the distance from the superconductor, which depends sensitively on the junction orientation; (ii) In planar Josephson junctions, although there is no net magnetization, 0-$\pi$ transitions occur as a function of

**Table 1 | Comparison of different approaches to generate finite-momentum Cooper pairing**

| Systems | Rashba[55] | Dirac surface states[18,19] | Altermagnet (this work) |
|---|---|---|---|
| Finite momentum $q$ | $H_\parallel / \sqrt{\alpha_R^2 + \mu^2/m}$ | $H_\parallel / v_F$ | $\sqrt{2\mu}\sin(2\theta)$ |
| Magnetic field (magnetization) | in-plane field | in-plane field | no field |
| $\theta$ dependence | no | no | yes |
| Large oblique-angle transport | no | no | yes |
| $\mu$ dependence | large $\mu$, smaller $q$ | no | larger $\mu$, larger $q$ |

It summarizes theoretical results (e.g., finite momentum $q$, requirements of magnetic fields (magnetization), dependence on the chemical potential $\mu$ and propagation direction $\theta$, and large oblique-angle transport) for three systems: (1) Rashba superconductor with $s$-wave pairing under in-plane magnetic field $H_\parallel$. $\alpha_R$ is the Rashba spin-orbit coupling, $\mu$ is the chemical potential and $m$ is the effective mass. (2) Dirac surface states with proximity-induced $s$-wave pairing under in-plane magnetic field $H_\parallel$. $v_F$ is the Fermi velocity. (3) Altermagnet with proximity-induced $s$-wave pairing.

gate voltage on the altermagnet, length or orientation of the junction; (iii) In junctions parallel to the direction where the spin splitting of the altermagnet vanishes, the superconducting transport is dominated by Cooper pairs moving at large oblique angles away from the junction direction; (iv) Josephson junctions oriented along different directions exhibit distinct Fraunhofer interference patterns in response to external magnetic fields. These results are generic and do not rely on the details of the $s$-wave superconductors and junction interfaces.

Compared to the previously studied platforms for realizing finite-momentum pairings, such as Rashba superconductors and Dirac surface states with proximity-induced superconductivity, our proposal does not require a magnetic field (or net magnetization). In the altermagnetic system, the Cooper-pair momentum depends substantially on the propagation direction $\theta$ and the larger oblique-angle Cooper transfer can dominate the superconducting transport, which does not exist in the ferromagnetic junctions. Additionally, the Cooper-pair momentum exhibits a rather different dependence on the chemical potential $\mu$ of the system. We summarize the comparisons between the typical ferromagnetic and altermagnetic systems in Table 1.

There has been a growing number of candidate materials predicted and confirmed as altermagnet. Among them, a prime example is the collinear $RuO_2$ that has been widely studied theoretically and experimentally[28,31,36,50–52]. $RuO_2$ processes a strong anisotropic spin-splitting (on the eV scale) in the electronic band structure, which has been verified in ARPES experiment recently[53]. Thin films of $RuO_2$ have also been realized and, interestingly, with signatures of intrinsic superconductivity[51,52]. Using typical parameters for this material, $Ja^2 = 1$ eV, $ta^2 = 2.5$ eV, lattice constant $a = 4.5$ Å and doping $\mu = 0.3–0.5$ eV[28,35], we estimate the shortest periodicity (at $\varphi = 0$) with respect to the junction length as $\mathcal{P}_y = 2\pi\sqrt{t^3/(\mu J^2)} \sim 16–20$ nm. Thus, it is feasible to observe the oscillations of order parameter and Josephson current in superconducting junctions with lengths longer than 20 nm. Similar to $RuO_2$, $KRu_4O_8$ has a $d$-wave-like altermagnetic order but is described by $J(k_x^2 - k_y^2)s_z/2$[31]. It is precisely related to $Jk_x k_y s_z$ by 45-degree rotation about $z$-axis. The spin splitting maximizes in the $k_x$- and $k_y$-axes while vanishes in the $k_x = \pm k_y$ directions. Thus, our results also apply to this material but with swapping the directions parallel and diagonal to the crystal axis. For $KRu_4O_8$, we take the parameters $ta^2 = 0.05$ eV, $Ja^2 = 0.037$ eV, $a = 9.9$ Å and $\mu = 0.03–0.1$ eV[31] and estimate the shortest periodicity (at $\varphi = \pi/4$) a $\mathcal{P}_y \sim 6–11$ nm. Therefore, the oscillations could be observed in $KRu_4O_8$-based junctions with shorter lengths compared to those of $RuO_2$.

So far, we have focused on the altermagnet metals with $d$-wave magnetism. Our main results, however, can be easily generalized to other planar altermagnetic metals, e.g., with $g$- or $i$-wave magnetism. In particular, we expect pronounced oscillations in the induced order parameter with respect to junction length or doping for any junction orientation. The periodicity of the oscillations is determined by the junction orientation. It is the largest (smallest) when the junction is aligned with the direction where the spin splitting of the altermagnet vanishes (maximizes). In Josephson junctions, anomalous 0-$\pi$

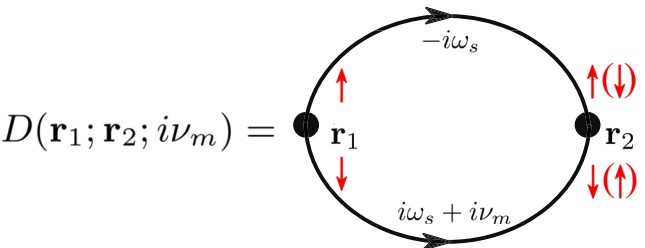

**Fig. 6 | Diagram for the propagator for an electron pair.** The vertex correction due to the scattering of the electrons is not considered for simplicity. The red arrows indicate opposite spins carried by the electron pair.

transitions occur despite the zero net magnetization. Moreover, the superconducting transport is dominated by Cooper pairs moving at large oblique angles away from the junction direction when the junction is in the direction where the spin splitting vanishes. Thus, the current density becomes sensitive to the width of the junction.

Our work not only sheds light on the exploration and understanding of finite-momentum Cooper pairing in the absence of net magnetization but also uncovers exotic superconducting phenomena in the altermagnet. While our focus lies on the most typical scenario where the altermagnet has the Fermi surfaces at the $\Gamma$ point and the superconductivity of $s$-wave singlet pairing, it would be interesting to extend our study to the case with valley degrees of freedom in the altermagnet[32] or involving triplet pairing, in particular, in the presence of spin-orbit coupling. It would also be promising to explore topological superconductivity and nonreciprocal superconducting phenomena, such as the diode effect, in altermagnets without net magnetization.

Recently, we became aware of a numerical study[54] which discusses 0-$\pi$ Josephson transitions.

## Methods
### Cooper-pair propagator
The Cooper-pair propagator describes the simultaneous propagation of two electrons with opposite spins initially from a position $\mathbf{r}_1$ at time $t_1$ to another position $\mathbf{r}_2$ at time $t_2$. Using Wick's theorem and the Matsubara formalism, we can evaluate it as a convolution of two electron Green's functions

$$D(\mathbf{r}_1; \mathbf{r}_2; i\nu_m) = \frac{1}{2\beta}\sum_{\omega_s} \text{Tr}\left[\mathcal{G}_0(\mathbf{r}_1, \mathbf{r}_2, -i\omega_s)s_y\mathcal{G}_0^T(\mathbf{r}_1, \mathbf{r}_2, i\omega_s + i\nu_m)s_y\right], \quad (10)$$

where $\mathcal{G}_0(\mathbf{r}_1, \mathbf{r}_2, i\omega_s)$ is the non-interacting Matsubara Green's function. It is a $2 \times 2$ matrix in the spin basis $(\psi_\uparrow, \psi_\downarrow)$. $\omega_s = (2s+1)\pi k_B T$ and $\nu_m = 2\pi m k_B T$ (with $s$ and $m$ being integers) are Matsubara frequencies for fermions and bosons (here electron pairs). The Feynmann diagram for the pair propagator is shown in Fig. 6. Assuming translation symmetry and considering the static limit, the Cooper pair propagator

at zero temperature can be found as

$$D(\mathbf{r}) = D(\mathbf{r}, 0) = \frac{1}{2} \int_0^\infty \frac{d\epsilon}{2\pi} \frac{d\epsilon'}{2\pi} \frac{\mathcal{T}(\mathbf{r}, \epsilon, \epsilon') + \mathcal{T}(\mathbf{r}, -\epsilon, -\epsilon')}{\epsilon + \epsilon'}. \quad (11)$$

where $\mathbf{r} = \mathbf{r}_2 - \mathbf{r}_1$,

$$\mathcal{T}(\mathbf{r}, \epsilon, \epsilon') = \mathrm{Tr}\left[g_0(\mathbf{r}, \epsilon) s_y g_0^T(\mathbf{r}, \epsilon') s_y\right], \quad (12)$$

and $g_0(\mathbf{r}, \epsilon)$ is the Fourier transform of the spectral function matrix given by

$$g_0(\mathbf{r}, \epsilon) = \int \frac{d^2\mathbf{k}}{(2\pi)^2} e^{i\mathbf{k}\cdot\mathbf{r}} A_0(\mathbf{k}, \epsilon). \quad (13)$$

For the altermagnet, the spectral function matrix is obtained from the Green's function as

$$A_0(\mathbf{k}, \epsilon) \equiv -2\,\mathrm{Im}\left[G_0^{\mathrm{ret}}(\mathbf{k}, \epsilon)\right]$$
$$= 2\pi \sum_{\eta = \pm 1} \delta(\epsilon - \varepsilon_{\mathbf{k}, \eta}) \frac{1 + \eta s_z}{2}, \quad (14)$$

where $\eta = \pm 1$ distinguishes the two bands $\varepsilon_{\mathbf{k}, \eta}$ of opposite spins.

## Local order parameter

We derive the order parameter $\langle |\Psi(\mathbf{r}')| \rangle$ by integrating Eq. (4) in the main text. When the interface (junction) width and chemical potential in the altermagnet are large, i.e., $W \gg y' \gg 1/(J\sqrt{\mu})$, the local order parameter $\langle |\Psi(\mathbf{r}')| \rangle$ (not close to the edges at $x' = \pm W/2$) becomes independent of $W$ and position $x'$ along the interface. This result holds for interfaces in any orientation. Thus, it suffices to calculate the order parameter at $x' = 0$ which reads

$$\langle |\Psi(0, y')| \rangle$$
$$= \frac{2\lambda}{\pi^2} \int_{-W/2}^{W/2} dx_1' \frac{(J_-' J_+')^{3/2}}{r'^2(J_+' + J_-')} \cos\left[\sqrt{2\mu} r'(J_-' - J_+')\right], \quad (15)$$

where $J_\pm' = 1/\sqrt{2 \pm J\sin(2\theta' + 2\varphi)}$. Converting the integral over $x'$ to an integral over the angle $\theta'$ (i.e., $dx' = -\csc^2\theta' d\theta'$), we have

$$\langle |\Psi(0, y')| \rangle = \frac{2\lambda}{\pi^2} \frac{1}{y'} \int_\alpha^{\pi - \alpha} d\theta' \frac{(J_+' J_-')^{3/2}}{J_+' + J_-'} \cos(\sqrt{\mu} \mathcal{F}_{\theta'} y'), \quad (16)$$

where $\alpha = \arctan(2y'/W)$ and $\mathcal{F}_{\theta'} = \sqrt{2}\csc\theta'(J_-' - J_+')$. For large $y' \gg 1/J\sqrt{\mu}$, the function $\cos(\sqrt{\mu} y' \mathcal{F}_{\theta'})$ oscillates rapidly as $\theta'$ varies. Thus, we can apply the saddle point approximation to the integral over $\theta'$ and obtain the analytical result in Eq. (5).

## Josephson supercurrent

We derive the formula for Josephson supercurrent. The pairing interaction of the planar Josephson junction can be written as

$$H_p = -\int d^2\mathbf{r}'\left[\Delta(\mathbf{r}')\Psi^\dagger(\mathbf{r}') + \Delta^*(\mathbf{r}')\Psi(\mathbf{r}')\right], \quad (17)$$

where $\Psi(\mathbf{r}') \equiv \psi_\uparrow(\mathbf{r}')\psi_\downarrow(\mathbf{r}')$ and $\Delta(\mathbf{r}')$ is the pairing potential. We assume

$$\Delta(\mathbf{r}') = \lambda_1(x')\delta(y') + \lambda_2(x')\delta(y' - L), \quad (18)$$

with $|x'| < W/2$ and that the magnitude of $\lambda_j > 0$ (with $j \in \{1, 2\}$) is constant along the superconducting lead, $\lambda_j(x') = \lambda_j e^{i\phi_j}$.

The supercurrent is given by the derivative of the free energy with respect to the pairing phase difference $\delta_\phi = \phi_1 - \phi_2$ across the junction. The free energy contributed by the pairing interaction, $F_p = \langle |H_p| \rangle$,

reads

$$F_p = -\int dx'\left[\lambda_1(x')\langle |\Psi^\dagger(x', 0)| \rangle + \lambda_2(x')\langle |\Psi^\dagger(x', L)| \rangle\right.$$
$$\left. + \lambda_1^*(x')\langle |\Psi(x', 0)| \rangle + \lambda_2^*(x')\langle |\Psi(x', L)| \rangle\right]. \quad (19)$$

Here, $\langle |\Psi(\mathbf{r}')| \rangle$ is the local order parameter, induced from the two superconducting leads by proximity effect. It can be written as $\langle |\Psi(\mathbf{r}')| \rangle = \langle |\Psi(\mathbf{r}')| \rangle_1 + \langle |\Psi(\mathbf{r}')| \rangle_2$ with

$$\langle |\Psi(\mathbf{r}')| \rangle_1 = \int dx_1' \lambda_1(x_1') D(\mathbf{r}'; x_1', 0),$$
$$\langle |\Psi(\mathbf{r}')| \rangle_2 = \int dx_1' \lambda_2(x_1') D(\mathbf{r}'; x_1', W). \quad (20)$$

Plugging Eq. (20) into Eq. (19), the supercurrent $I_s \equiv (e/\hbar)\partial F_p/\partial\delta_\phi$ is found as

$$I_s(\delta_\phi) = -i\frac{2e}{\hbar}\lambda_1\lambda_2 \int_{-W/2}^{W/2} dx' dx_1'$$
$$\times [e^{i\delta_\phi} D(x', L; x_1', 0) - e^{-i\delta_\phi} D(x', 0; x_1', L)] \quad (21)$$

Exchanging dummy variables, we replace $D(x', 0; x_1', L)$ by $D(x', L; x_1', 0)$. Using the fact that the propagator is real-valued, we derive eventually $I(\delta_\phi) = I_c \sin(\delta_\phi)$ with the critical supercurrent given by Eq. (8).

## Data availability

The datasets generated during this study are available from the corresponding authors upon request.

## Code availability

The custom codes generated during this study are available from the corresponding authors upon request.

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

## Acknowledgements

We thank Jian Li and Zhe Yuan for helpful discussion. This work was partially carried out at the University of Science and Technology of China (USTC) and Hefei National Laboratory (HFNL). S.-B.Z. acknowledges the support of the start-up funds at HFNL and USTC. L.-H.H. at Aalto is funded by the Jane and Aatos Erkko Foundation and the Keele Foundation as part of the SuperC collaboration. The work was also supported by the European Research Council (ERC) under the European Union's Horizon 2020 research and innovation program (ERC-StGNeupert-757867-PARATOP), the Innovation Program for Quantum Science and Technology (Grant No. 2021ZD0302800), and Anhui Initiative in Quantum Information Technologies (Grant No. AHY170000).

## Author contributions

S.-B.Z. and L.-H.H. conceived the project idea. S.-B.Z. performed the calculations and wrote the manuscript with input from L.-H.H. and T.N. All authors contributed to the scientific discussions and manuscript revisions.

## Competing interests

The authors declare no competing interests.
