## [Peer Review File · Nature Communications]

Finite-momentum Cooper pairing in proximitized altermagnetsREVIEWER COMMENTS

Reviewer #1 (Remarks to the Author):

Zhang et al. provide a theoretical demonstration of Cooper pairing with finite angular momentum in altermagnetics, the recently discovered unconventional d, g, I-wave magnets. Cooper pairing with finite angular momentum is usually associated with magnetization, but in this study the authors remove this limitation by demonstrating the pairing in altermagnetic systems with zero net magnetization. The authors also study the angular dependence of the Josephson oscillations in an altermagnet from an adjacent s-wave superconductor, and find characteristic anisotropic properties of the oscillations and the envelope function-both of which depend strongly on whether they are probed along the direction of the altermagnetically split or spin-degenerate wavefunctions (corresponding to the d-wave ordering parameter). The authors also make estimates of the characteristic parameters of the Josephson transitions based on available DFT calculations of the altermagnetic candidates RuO₂ and KRuO₄.

The authors address the timely topic of the interaction of the emerging magnetic class of altermagnetism with superconductivity, which is of potential interest across various communities in condensed matter physics, as evidenced by the growing number of papers on the interaction of superconductors and altermagnets on the arxiv. The authors also convincingly demonstrate the unique angular dependence of Josephson effects in altermagnets and contrast its properties with properties of more conventional superconducting junctions based on ferromagnetic, Rashba or Dirac systems. Thus, in our opinion, the results fit well with Nature Communications.

Before publishing the manuscript, we would recommend reviewing the following few points that might clarify some aspects and improve the readability of the manuscript for non-experts.

- In the introduction, the authors could explain what $(0, \pi)$ means for non-experts.
- In the section on material candidates - it might be clearer to list the order parameters for RuO₂ and KRuO₄ as $k_x k_y s_z$ and $k_x^2 - k_y^2$ instead of $k_x^2 - k_y^2 s_z$ and $k_x k_y s_z$. (It may be still noted - as in the current version - that the two types are related precisely by the 45 degree rotation of the axes.) In this way, the long-wavelength Hamiltonians correspond directly to the order parameters in Table of Reference 22.
- When the authors talk through the manuscript about the C₄ symmetry, isn't the symmetry in fact nonrelativistic spin symmetry? For the studied Hamiltonians do not have spin-orbit coupling. Again, clarifying this could make cleaner link to the existing references.
- When the long-wavelength altermagnetic model $k_x k_y s_z$ is referenced; also Ref. 27 could be cited.
- Could the authors speculate how the conclusion are changed when the spin-orbit interaction is taken into account?

Reviewer #2 (Remarks to the Author):

In this work, authors study the superconductivity induced from a conventional BCS superconductor into an altermagnet metallic material. They study the formation of finite-momentum Cooper pairs in the altermagnet in the absence of a net magnetization. The superconducting order parameter decays into the superconductor in an oscillating way that also depends on the doping of the metal. Depending on the relative orientation between the altermagnet crystalline axis and the superconductor, they found a Cooper pair propagation that can go predominantly from zero to large oblique angles. Overall, I think the article contains relevant information for the field and is well-written. Also, the methodology seems to be in general appropriate for the problem they are looking at.

Although the results are new and deserve to be published, I am not convinced this work defines the advancement needed to recommend for publication in Nature Communications. For instance, I find lacking an experiment proposal where their main predictions can be tested: Authors spend a great part of the paper explaining the dependence of certain quantities on the angle between the altermagnet's crystalline axis and the interface with the superconductor. However, they don't propose a way to tune this angle and, therefore, probe the unique properties emerging from the "d-wave magnetism". While observations including alternating 0- π transitions as a function of the length can be observed in other materials, including ferromagnets.

I have some minor questions as well:

1. Can authors provide an intuitive explanation about why the superconducting pairing depends on the chemical potential? If the ferromagnet is a metal, I would naively expect properties to be rather independent from μ .
2. It seems that all properties are $\pi/2$ periodic due to the lattice symmetry. However, panel 1(c) is not $\pi/2$ periodic. Can authors clarify the reason why this is the case?
3. In the first paragraph of page 3, authors claim: "This suggests that the measurement of I_c gives direct access to information about the order parameter." However, I don't see how this is the case. If we have an altermagnet with a thickness L embedded in a Josephson junction, the maximum distance between a point of the altermagnet and one of the superconductors is $L/2$. It is remarkable that Eqs. (5) and (9) have the same functional form, but the interpretation from the authors seems a bit misleading.
4. Authors make a comment about the difference between altermagnetic and anti-ferromagnetic materials. Can they elaborate more on it? I think this distinction might be important to distinguish between the two scenarios.
5. In the last paragraph before conclusions, authors say that they use numerical calculations to test transport through a finite-size junction which might be a promising measurement to probe that Cooper pairs move at large oblique angles. However, no details are given about the numerical calculations in the main text. The text will benefit from a brief discussion of the methods.
5. In the same paragraph, authors claim that oscillations are strongly suppressed for $W < L$. While it is easy to understand that the Cooper pairs traveling in the main direction ($\theta = \pi/2$) do not reach directly the superconductor on the other side, one would naively expect a perfect reflection at the boundary of the altermagnet would allow the pair to reach the other end. Is this picture wrong? Why are the Cooper pairs reaching the boundaries with vacuum lost?
6. Can authors comment about the possibility of measuring the $W < L$ regime using RuO₂ (or other material)? The resolution of the features proposed (10 nm or less) seems challenging from an experimental fabrication perspective.

Reviewer #3 (Remarks to the Author):

The manuscript by Song-Bo Zhang et al. reports on the theoretical study of Cooper pairing in an altermagnetic metal in proximity to conventional s-wave superconductors. Many interesting results are reported, in particular concerning Josephson junctions composed of an altermagnetic metal as barrier and conventional s-wave superconducting electrodes. Specifically, oscillations of the damping parameter and 0- π transitions occur by varying different parameters of the altermagnet barrier, even in absence of a net magnetization. Moreover, the transport properties are anisotropic, reflecting in some sense the anisotropic features of the altermagnet barrier.

In my opinion, the topic of the manuscript is timely and quite interesting, the analysis is well supported and the paper is well written. Therefore, I recommend publication of the manuscript in Nature Communications, after a few remarks have been considered.

Specifically, the transport properties strongly depend on the interface orientation between the superconducting electrodes and the altermagnet barrier. It turns out that some crucial unconventional behaviours can be observed if there is control on the interface orientation. For instance, the reduction of the critical current density, when reducing the width of the junction, can be observed when the junction is along the direction where the spin splitting vanishes.

As widely reported in the huge literature on anisotropic HTS superconductors and HTS junctions, control on junction's interface can be quite difficult to achieve. Moreover, non-uniformity in the barrier along the junction width can induce a mixing of the transport properties, since the interface direction is not the same along the width. For these reasons, some of the experiments proposed by the Authors can be hard to provide clear results. In this sense, phase sensitive experiments can be more powerful, as for HTS junctions. Therefore, it would be very useful if the Authors can provide predictions concerning the behaviour of the mentioned junctions in presence of magnetic field, the magnetic field pattern of the critical current, or the behaviour in a SQUID geometry. This would provide much more powerful insights to be tested in an experiment.

Response to the Reviewers

Nature Communications NCOMMS-23-33963A

Reply to Review #1

[General comments]: Zhang et al. provide a theoretical demonstration of Cooper pairing with finite angular momentum in altermagnetics, the recently discovered unconventional d, g, I-wave magnets. Cooper pairing with finite angular momentum is usually associated with magnetization, but in this study the authors remove this limitation by demonstrating the pairing in altermagnetic systems with zero net magnetization. The authors also study the angular dependence of the Josephson oscillations in an altermagnet from an adjacent s-wave superconductor, and find characteristic anisotropic properties of the oscillations and the envelope function-both of which depend strongly on whether they are probed along the direction of the altermagnetically split or spin-degenerate wavefunctions (corresponding to the d-wave ordering parameter). The authors also make estimates of the characteristic parameters of the Josephson transitions based on available DFT calculations of the altermagnetic candidates RuO₂ and KRuO₄.

The authors address the timely topic of the interaction of the emerging magnetic class of altermagnetism with superconductivity, which is of potential interest across various communities in condensed matter physics, as evidenced by the growing number of papers on the interaction of superconductors and altermagnets on the arxiv. The authors also convincingly demonstrate the unique angular dependence of Josephson effects in altermagnets and contrast its properties with properties of more conventional superconducting junctions based on ferromagnetic, Rashba or Dirac systems. Thus, in our opinion, the results fit well with Nature Communications.

Before publishing the manuscript, we would recommend reviewing the following few points that might clarify some aspects and improve the readability of the manuscript for non-experts.

[Reply]: We are greatly encouraged by the reviewer's positive assessment and appreciate the insightful and constructive comments. We also thank the reviewer for the support in publishing our manuscript in Nature Communications. Below is our one-to-one response.

[Comment 1]: - In the introduction, the authors could explain what $(0, \pi)$ means for non-experts.

[Reply 1]: We thank the reviewer for this constructive suggestion. The 0 and π states of Josephson junction can be elucidated by plotting the free energy of the junction as a function of pairing phase difference across the junction. For an ordinary junction (the 0-junction), the

ground state (with the lowest free energy) occurs when the phase difference is zero. In contrast, for a π -junction, the ground state occurs instead at π phase difference.

Following the reviewer's suggestion, we have added the explanation "*In Josephson junctions, the ground state usually has no phase difference across the junction and is referred to a 0-junction. However, the finite magnetization may produce an intrinsic π phase difference, forming a so-called π -junction [17,18]. Notably, a switchable π state of the Josephson junction holds important applications in superconducting circuits and qubits [19-21]*" to the end of the first paragraph of the revised manuscript.

[Comment 2]: - In the section on material candidates - it might be clearer to list the order parameters for RuO₂ and KRuO₄ as $k_x k_y s_z$ and $k_x^2 - k_y^2$ instead of $k_x^2 - k_y^2$ s_z and $k_x k_y$ s_z . (It may be still noted as in the current version - that the two types are related precisely by the 45 degree rotation of the axes.) In this way, the long-wavelength Hamiltonians correspond directly to the order parameters in Table of Reference 22.

[Reply 2]: We thank the reviewer for this constructive suggestion. Following the reviewer's suggestion, we have corrected the order parameters of RuO₂ and K₂RuO₈ as $k_x k_y s_z$ and $(k_x^2 - k_y^2) s_z$ and referred to Ref. [23] explicitly in the revised manuscript. We have also mentioned explicitly that these two types of altermagnetic order are related to each other precisely by an rotation of 45 degrees about z axis.

[Comment 3]: - When the authors talk through the manuscript about the C₄ symmetry, isn't the symmetry in fact nonrelativistic spin symmetry? For the studied Hamiltonians do not have spin-orbit coupling. Again, clarifying this could make cleaner link to the existing references.

[Reply 3]: The reviewer is correct, the C₄ symmetry we used is indeed nonrelativistic or spinless. In the previous version, we used the magnetic space group to explain the symmetry of the Hamiltonian, i.e., C_4T , a four-fold rotation followed by time-reversal symmetry. In the revised manuscript, we have followed Ref. 23 and used alternatively the spin-space group to explain the symmetry of the *d*-wave altermagnetic Hamiltonian as $[C_2||C_{4z}]$, i.e., a four-fold rotation in real space $(x, y) \rightarrow (y, -x)$ together with a two-fold rotation in spin space $(\uparrow, \downarrow) \rightarrow (\downarrow, \uparrow)$. For the long wavelength Hamiltonian, these two descriptions are the equivalent.

In the revised manuscript, we have rephrased this description as "*The model respects $[C_2||C_{4z}]$ symmetry, i.e., a four-fold rotation in real space $(x, y) \rightarrow (y, -x)$ together with a two-fold rotation in spin space $(\uparrow, \downarrow) \rightarrow (\downarrow, \uparrow)$, which is indicated by the relation $s_y \mathcal{H}^*(k_x, k_{xy}) s_y = \mathcal{H}(k_y, -k_x)$.*"

[Comment 4]: - When the long-wavelength altermagnetic model $k_x k_y s_z$ is referenced; also Ref. 27 could be cited.

[Reply 4]: We have added Ref. 27 as a reference to Eq. (1) in the revised manuscript.

[Comment 5]: - Could the authors speculate how the conclusion are changed when the spin-orbit interaction is taken into account?

[Reply 5]: We thank the reviewer for raising this question. In the current manuscript, we do not consider spin-orbit coupling because altermagnetic materials, such as RuO₂ and KRu₄O₈, are usually described by spin symmetry group, which is possible precisely because spin-orbit coupling is neglected. In these materials, the energy scale of spin-orbit coupling is much smaller than the normal kinetic energy and altermagnetic exchange energy. In the absence of spin-orbit coupling, the two bands of opposite spin cross in specific directions (e.g., k_x and k_y axes). Spin-orbit coupling (e.g., the Rashba-type described by $k_x s_y - k_y s_x$) would avoid such band crossing and cause an in-plane spin texture around the avoided crossing points (see Fig. R1 below). However, without an external magnetic field, there is still no Cooper-pair momentum in these directions (i.e., x or y directions). Note that in this direction, the model resembles that for nonmagnetic metals. Thus, the Cooper-pair momentum \vec{q} should still take the form $\sim \sin 2\theta_k$, as enforced by the four-fold rotation symmetry. However, the magnitude of $|\vec{q}|$ may be some tiny corrections by spin-orbit coupling. Moreover, due to the spin rotation caused by spin-orbit coupling (i.e., inversion symmetry is broken), even-parity pairing could coexist with odd-parity pairing. Thus, spin-triplet Cooper pairing may be induced in the system.

However, since spin-orbit coupling follows time-reversal symmetry and does not introduce any net magnetization, the inclusion of small spin-orbit coupling would not qualitatively change our main conclusions such as the existence of finite-momentum Cooper pairing and $0-\pi$ Josephson transition in absence of a net magnetization. As we discuss in the manuscript, all anisotropic phenomena essentially stem from the unique direction-dependence of Cooper pair momentum ($q \sim \sin 2\theta_k$). In particular, in the direction where the Fermi surfaces are most split (e.g., the $k_x = k_y$ direction), the Fermi surfaces are barely altered by spin-orbit coupling. Therefore, we expect that the momentum of Cooper pairs propagating in this direction and the superconducting transport properties in junctions oriented along this direction stay almost the same. The influence of moderate and strong spin-orbit coupling is an interesting question and deserves an in-depth and systematic study. However, it goes beyond the scope of the current work. We defer it to a further study.

In the revised manuscript, we added a brief comment on spin-orbit coupling in the discussion section “*it would be interesting to extend our study to the case with valley degrees of freedom in the altermagnet or involving triplet pairing, in particular, in the presence of spin-orbit coupling.*”

Fig. R1. (a) Sketch of the spin-split Fermi surfaces in the absence of spin-orbit coupling. (b) Sketch of the spin-split Fermi surfaces in the presence of a small Rashba-type spin-orbit coupling. Red and blue colors indicate positive and negative spin polarization, respectively.

 Reply to Review #2

[General comments]: In this work, authors study the superconductivity induced from a conventional BCS superconductor into an altermagnet metallic material. They study the formation of finite-momentum Cooper pairs in the altermagnet in the absence of a net magnetization. The superconducting order parameter decays into the superconductor in an oscillating way that also depends on the doping of the metal. Depending on the relative orientation between the altermagnet crystalline axis and the superconductor, they found a Cooper pair propagation that can go predominantly from zero to large oblique angles. Overall, I think the article contains relevant information for the field and is well-written. Also, the methodology seems to be in general appropriate for the problem they are looking at.

[Reply]: We thank the reviewer for the careful reading and concise summary of our work. We are also grateful to the reviewer for the insightful and constructive comments and criticisms, which indeed have motivated us to improve the manuscript substantially.

Although the results are new and deserve to be published, I am not convinced this work defines the advancement needed to recommend for publication in Nature Communications. For instance, I find lacking an experiment proposal where their main predictions can be tested: Authors spend a great part of the paper explaining the dependence of certain quantities on the angle between the altermagnet's crystalline axis and the interface with the superconductor. However, they don't propose a way to tune this angle and, therefore, probe the unique properties emerging from the "d-wave magnetism". While observations including alternating

0- π transitions as a function of the length can be observed in other materials, including ferromagnets.

[Reply]: We thank the reviewer for this criticism. However, we have a different perspective on the advancement of our work needed for Nature Communications. First, while it was widely believed that finite-momentum Cooper pairing requires a *finite* net magnetization, in the manuscript, we challenge this belief by revealing that magnetic systems with *zero* net magnetism can after all support Cooper pairs carrying finite momentum. To demonstrate this, we take altermagnets as an example and systematically study superconducting proximity effect in the altermagnet close to *s*-wave superconductors. Strikingly, we have found that despite *zero* net magnetization in the system, the induced Cooper pairs acquire a finite center-of-mass momentum when propagating in the altermagnet. We have also revealed that the finite Cooper-pair momentum strongly depends on the propagation direction and exhibits a unique anisotropic symmetric pattern, in stark contrast to previously known magnetic systems.

Furthermore, we have shown that this anomalous finite-momentum pairing can manifest as several novel and measurable phenomena, such as (i) highly orientation-dependent oscillations in the order parameter, (ii) 0- π transitions in Josephson junctions that is highly tunable by modulating doping, length, or orientation of the altermagnet, and (iii) large-oblique-angle dominated Cooper pair transport in junctions parallel to a crystalline axis. Note that all these features occur in the absence of a net magnetization, in sharp contrast to the ferromagnetic and antiferromagnetic systems. And to the best of our knowledge, feature (iii) has never been seen in any previously studied system.

Based on these discoveries, we are convinced that our work not only sheds light on the exploration and understanding of finite-momentum Cooper pairing in the absence of a net magnetization, but also represents an important step forward in understanding the interplay of magnetic orders and superconductivity in proximitized materials. Importantly, our results can be implemented in many candidate materials of altermagnets, some of which have been actively studied in recent years, such as RuO₂, KRu₄O₈, and Mn₅Si₃. Therefore, we strongly believe that our predictions can be tested experimentally in the near future, and our work will be of broad and immediate interest to theorists and experimentalists across the wide fields of superconductivity and magnetism.

We agree with the reviewer that finite-momentum Cooper pairing and 0- π transition by tuning the junction length have also been observed in ferromagnetic systems. In the manuscript, we have compared our results with typical proposals requiring magnetization (see Table. 1 in the manuscript), which clearly demonstrates the sharp differences between our proposal with the others. In addition to tuning the junction length, we have also shown 0- π transitions by alternatively tuning the doping in the altermagnet or the junction orientation.

We also agree with the reviewer that it might be challenging to rotate the junction orientation in experiments. However, this is not impossible. In fact, using photolithography and ion milling techniques, it is feasible to fabricate curved devices with multiple accurate leads that can

effectively vary the junction orientation continuously (i.e., by changing the lead pairs at different positions), see a sketch in Fig. R2 below. Notably, curved devices have been recently developed and applied successfully to probe the anisotropic properties in magnetic materials, see e.g., Phys. Rev. Lett. **128**, 247202 (2022) and Adv. Electron. Mater. **9**, 2300049 (2023). By replacing the normal metallic leads with conventional superconducting leads such as aluminium and niobium, it is possible to achieve the $0\text{-}\pi$ transition by effectively varying the junction orientation.

Fig. R2. Sketch of a curved device. Changing the pairs of leads at different positions, this allows an effective rotation of the junction orientation. Adapted from Adv. Electron. Mater. **9**, 2300049 (2023).

Motivated by the reviewers' comments, we have further investigated the Fraunhofer interference pattern in response to an external magnetic field. Interestingly, we find that the junctions oriented along the maximal and vanishing spin-splitting directions exhibit distinct Fraunhofer patterns due to their different dominant Cooper-pair transfer trajectories. In particular, at the $0\text{-}\pi$ transition point of the junction along the direction where spin-splitting vanishes, the critical supercurrent can be induced and enhanced by applying a magnetic field (see Fig. R6 below). In contrast, in the junction oriented along the direction of maximal splitting, we observe a normal Fraunhofer pattern, in which the maximum critical supercurrent is at zero field. This contrast provides us with another signature to detect the unique superconducting transport properties of altermagnetic junctions. Note that Fraunhofer patterns can be measured directly in experiments.

In the revised manuscript, we have added a sentence “*Experimentally, one could fabricate curved devices with a series of superconducting lead pairs, similar to those used for anisotropic magnetoresistance measurements [41,42], which allows an effective rotation of the junction orientation*” and added the new references [41,42] to point out the feasibility of effectively varying the junction orientation. We have also provided a new section and a new figure 5 to discuss the results of Fraunhofer pattern and added the relevant calculation to the Supplementary Information (Sec. IV “Calculations of the Fraunhofer pattern”).

Below are our one-to-one responses to your additional comments. With these improvements, we hope that the reviewer could agree with us that the revised manuscript should be published in Nature Communications.

[Comment 1]: 1. Can authors provide an intuitive explanation about why the superconducting pairing depends on the chemical potential? If the ferromagnet is a metal, I would naively expect properties to be rather independent from μ .

[Reply 1]: We thank the reviewer for raising this question. An intuitive picture can be made by considering, e.g., the altermagnetic order described by $(k_x^2 - k_y^2)s_z$ (such as in KRu_4O_8) and looking the k_x axis. In this case, the Fermi momenta for the spin-up and spin-down electrons can be found by solving the equation $tk_x^2 \pm Jk_x^2 = \mu$, where μ is the chemical potential and \pm indicate the two spins. Specifically, we can find the Fermi momentum for the spin-up electron explicitly as $k_F^\uparrow = \pm\sqrt{\mu/(t+J)}$, while for the spin-down electron $k_F^\downarrow = \pm\sqrt{\mu/(t-J)}$. The Cooper pair consists of a spin-up electron with momentum $k_F^\uparrow = \sqrt{\mu/(t+J)}$ and a spin-down electron at $k_F^\downarrow = -\sqrt{\mu/(t-J)}$. Therefore, in the center of mass frame, the total momentum of the two electrons $q = \sqrt{\mu}(1/\sqrt{t-J} - 1/\sqrt{t+J})$ gives roughly the finite momentum of the Cooper pair $c_{\uparrow,k}c_{\downarrow,-k+q}$. In this way, it is obvious that the Cooper pairs induced by the proximity effect carry nonzero momentum q that depends crucially on the chemical potential μ . For a larger chemical potential μ , a larger Fermi surface splitting and hence a larger Cooper-pair momentum can be achieved.

[Comment 2]: 2. It seems that all properties are $\pi/2$ periodic due to the lattice symmetry. However, panel 1(c) is not $\pi/2$ periodic. Can authors clarify the reason why this is the case?

[Reply 2]: We thank the reviewer for this insightful comment. In fact, our analysis is based on the $k \cdot p$ Hamiltonian [cf. Eq. (1) in the main text] in the long wavelength limit. The model preserves a continuous rotation symmetry rather than crystalline rotation symmetry. For a typical ferromagnetic system with constant magnetism, this gives rise to a constant periodicity, as shown in Fig. 1 (e) in the manuscript. In contrast, the altermagnet exhibits strong anisotropy with $\pi/2$ periodicity as illustrated in Fig. 1(f), which is caused by the d -wave-like altermagnetism. For example, the periodicity of I_c oscillations varies dramatically for junctions with different orientations, with sharp cusps appearing at specific orientations corresponding to the directions in which the Fermi surface spin splitting vanishes (i.e., $\varphi = n\pi/2$). We agree with the reviewer that in a square lattice, ferromagnets might also exhibit a $\pi/2$ periodicity due to the lattice crystalline symmetry. However, in this work we focus on the long wavelength and low-energy regime where the electronic structure preserves approximately a continuous rotation symmetry. In this case, in ferromagnetic systems, the periodicity due to the lattice effect is usually very small, and the change of the periodicity by changing the junction orientation are also smooth.

To avoid any possible confusion and to be more precise, we have added the sentence to the caption of the figure “*In the calculation for the ferromagnet, we replace the altermagnetic*

order with a constant magnetization in Eq. (1) and consider the long-wavelength limit with rotation symmetry in the normal kinetic energy part.”

[Comment 3]: 3. In the first paragraph of page 3, authors claim: “This suggests that the measurement of I_c gives direct access to the information about the order parameter.” However, I don’t see how this is the case. If we have an altermagnet with a thickness L embedded in a Josephson junction, the maximum distance between a point of the altermagnet and one of the superconductors is $L/2$. It is remarkable that Eqs. (5) and (9) have the same functional form, but the interpretation from the authors seems a bit misleading.

[Reply 3]: We thank the reviewer for pointing out this misleading passage. By this statement, we aim to point out that the dependence of the order parameter on the position (measured from the altermagnet/superconductor interface), chemical potential in the altermagnet and junction orientation can be inferred from the measurement of the Josephson supercurrent, as a function of junction length, chemical potential, and junction orientation. Note that it is not a local probe of gap function by means of scanning tunneling microscope.

To avoid being ambiguous, we have rephrased the sentence as “*This suggests that the measurement of the supercurrent I_c , as a function of junction length, chemical potential and junction orientation can give access to the information about the order parameter, such as its dependence on position (measured from the AM-SC interface), chemical potential and junction orientation*” in the revised manuscript.

[Comment 4]: 4. Authors make a comment about the difference between altermagnetic and anti-ferromagnetic materials. Can they elaborate more on it? I think this distinction might be important to distinguish between the two scenarios.

[Reply 4]: In fact, altermagnets and collinear antiferromagnets share many similarities. For example, they can both be partitioned into a spin-up and a spin-down sublattice, and have fully compensated magnetization enforced by symmetry (see e.g., Phys. Rev. X **12**, 040501 (2022)). Thus, altermagnets were previously coined an unconventional collinear antiferromagnet. The distinction between them lies essentially in what symmetry connects the two spin sublattices. In classical collinear antiferromagnets, it is the combined symmetry of translation and pure spin rotation that flips the spin direction. As a result, the system has a double degeneracy (the Kramers degeneracy) at arbitrary momentum so that it can be effectively viewed as a metal with translation and time-reversal symmetries. Thus, finite-momentum Cooper pairing cannot be achieved in the antiferromagnets with s-wave superconductivity. In contrast, in altermagnets, the combined symmetry of translation and pure spin rotation is broken. However, the two spin sublattices can be instead connected by a combined symmetry of rotation and pure spin rotation. Therefore, the system has a spin-split band structure of strong anisotropy, resulting in unique magnetic patterns such as d -wave magnetism. This unique spin splitting is the key to realising finite-momentum Cooper pairs and hence the $0-\pi$ transitions in planar Josephson junctions.

Following the reviewer's suggestion, we have added a sentence "*It breaks the combined symmetry of translation and C_2 spin rotation that flips the spin direction (which is required for classical collinear antiferromagnets), but preserves a combined symmetry of spatial rotation and C_2 spin rotation*" to distinguish the altermagnetic and anti-ferromagnetic materials to the introduction of the revised manuscript.

[Comment 5]: 5. In the last paragraph before conclusions, authors say that they use numerical calculations to test transport through a finite-size junction which might be a promising measurement to probe that Cooper pairs move at large oblique angles. However, no details are given about the numerical calculations in the main text. The text will benefit from a brief discussion of the methods.

[Reply 5]: We thank the reviewer for this constructive suggestion. In the numerical calculation, we perform the direct numerical integration over the space without doing any approximations. We have added the description of the calculation at the end of the section "... *our numerical calculations in Figs. 4(b) and 4(d), where we perform the direct integration of x' and x'_1 from $-W/2$ to $W/2$ in Eq. (9)*".

[Comment 6]: 5. In the same paragraph, authors claim that oscillations are strongly suppressed for $W < L$. While it is easy to understand that the Cooper pairs traveling in the main direction ($\theta = \pi/2$) do not reach directly the superconductor on the other side, one would naively expect a perfect reflection at the boundary of the altermagnet would allow the pair to reach the other end. Is this picture wrong? Why are the Cooper pairs reaching the boundaries with vacuum lost?

[Reply 6]: We thank the reviewer for the insightful comments. In the calculation, we ignore side boundary reflections for simplicity as well as when obtaining analytical expressions of the order parameter and Josephson current. This is justified, in particular for junctions with $W > L$, because the Cooper-pair propagator decays quickly with increasing the propagation length and that the propagation length of the Cooper-pair propagator with side reflections is generally much longer than that of directly connecting the initial and final positions. Therefore, we can anticipate that the contribution from side reflections will not qualitatively change our main results, even for W not very smaller than L .

To substantiate this conclusion, we may perform the calculation as following. In the presence of side edge reflections, the Cooper-pair propagator may be modified as

$$D_{tot}(x_2, L; x_1, 0) = D(\mathbf{r}) + D_L(x_2, L; x_1, 0) + D_R(x_2, L; x_1, 0), \quad (1)$$

where D_L and D_R represent the corrections by the reflections at $x = \pm W/2$, respectively. Suppose the system geometry is mirror symmetric with respect to the center $x = 0$, D_L and D_R are of the same form. Thus, it suffices to derive D_L . In general, D_L can be complicated because the Fermi surface is not isotropic and the scattering property at the side edges is unknown. However, for simplicity, we may consider a junction with $\varphi = \pi/4$ and assume that the electrons are specularly reflected at the edge only once and without flipping spin, as

sketched in Fig. R3. Similar to the case with translational invariance, we may write D_L in the form

$$D_L(x_2, L; x_1, 0) = \int_0^\infty d\epsilon d\epsilon' \frac{\text{Tr}[h(x_2, L; x_1, 0; \epsilon) s_y h^T(x_2, L; x_1, 0; \epsilon') s_y] + \text{Tr}[h(x_2, L; x_1, 0; -\epsilon) s_y h^T(x_2, L; x_1, 0; -\epsilon') s_y]}{2(\epsilon + \epsilon')}, \quad (2)$$

where $h(x_2, L; x_1, 0; \epsilon)$ is the spectral function that corresponds to the Green's function of an electron moving from an arbitrary point $(x_1, 0)$ to another arbitrary point (x_2, L) via the specular reflecting point at $(-W/2, y_r)$ with $y_r = L(x_1 + W/2)/(x_1 + x_2 + W)$. The spectral function may be calculated as

$$h(x_2, L; x_1, 0; \epsilon) = \frac{1}{2\pi} \sum_\eta \int_0^{2\pi} d\phi e^{i(k_{1\eta} s_1 + k_{2\eta} s_2) \cos(\phi - \theta)} \frac{1}{2[1 + \eta J \sin(2\phi)/2]} P_\eta,$$

where $\mathbf{s}_1 = (-W/2 - x_1, y_r)$ and $\mathbf{s}_2 = (W/2 + x_2, L - y_r)$, θ is the direction of \mathbf{s}_1 , and

$$k_{1\eta} = \sqrt{\frac{\epsilon + \mu}{1 + \eta J \sin(2\phi)/2}}, \quad k_{2\eta} = \sqrt{\frac{\epsilon + \mu}{1 + \eta J \sin(\pi - 2\phi)/2}}.$$

Here, we have made use of the fact that for the specular reflection, the angle between \mathbf{k}_1 and \mathbf{s}_1 equals to that between \mathbf{k}_2 and \mathbf{s}_2 , and transformed the variables in the integral to polar coordinates and integrated over the magnitude of momentum.

Again, we consider the large chemical potential case with $k_1 s_1 + k_2 s_2 \gg 1$ and apply the saddle point approximation to the integration over θ . This results in

$$h(x_2, L; x_1, 0; \epsilon) \approx \frac{1}{(2\pi s)^{1/2} \mu^{1/4}} \sum_\eta \frac{P_\eta}{[1 + \eta J \sin(2\theta)/2]^{3/4}} (e^{i\bar{k}_\eta s - i\pi/4} + e^{-i\bar{k}_\eta s + i\pi/4}), \quad (3)$$

where $s = s_1 + s_2$ is the total propagation length, and \bar{k}_η is given by

$$\bar{k}_\eta = \sqrt{\frac{2\mu}{2 + \eta J \sin(2\theta)}} \left(1 + \frac{\epsilon}{2\mu}\right).$$

Note that when the initial point $(x_1, 0)$ or final point (x_2, L) approach to the boundary at $x = -W/2$, the reflection point coincides the initial or final point. In this case, Eq. (3) recovers the result without side reflections. Plugging Eq. (3) into Eq. (2) and following the same procedures as for the propagator without side reflections, we obtain

$$D_L(x_2, L; x_1, 0) = \frac{2 \cos[\sqrt{2\mu}(J_+ - J_-)s]}{\pi^2 s^2 (J_+ + J_-) [4 - J^2 \sin^2(2\theta)]^{3/4}}.$$

Clearly, the propagator decay quadratically with the propagation length s . Thus, for $L \lesssim W$, the correction due to the Cooper pair propagators with an edge reflection would be much smaller compared to those without an edge reflection. We can derive D_R similarly.

To further confirm this, we plug $D_{\text{tot}} = D + D_L + D_R$ into the formula

$$I_c = \frac{8e}{\hbar} \lambda_1 \lambda_2 \int_{-W/2}^{W/2} dx' dx'_1 D_{\text{tot}}(x', L; x'_1, 0)$$

and calculate numerically the supercurrent in junctions with different sizes and compare the results with those without side edge reflections. As shown in Fig. R4, we see that for $W > L$, the corrections from side edge reflections become ignorable. For $W \lesssim L$, we can also see that the reflection does not qualitatively change the results.

In the revised manuscript, we have added a comment on the side boundary reflection “*Here, we ignore the correction from side boundary reflections. This is justified for $L \lesssim W$, as we show in the Supplementary Information [40]*” and the referred to the relevant calculation with side boundary reflections to the Supplementary Information (Sec. V “Correction by side boundary reflections”).

Fig. R3. The blue and red lines represent the fastest trajectories in the propagators $D(2,1)$ and $D_L(2,1)$, respectively. $D_L(2,1)$ contains one specular reflection by the left edge boundary.

Fig. R4. Critical supercurrent density $I_c L/W$ (in units of $8e\lambda_1\lambda_2/\pi^2\hbar$) as a function of chemical potential μ for $\varphi = \pi/4$. For illustration, we consider $(W, L) = (1000, 20)$, $(W, L) = (100, 100)$ and $(W, L) = (100, 50)$ for (a), (b) and (c), respectively. The blue solid

and purple dash curves present the results without and with side edge reflections, respectively. For junctions with $L \lesssim W$, the corrections from side edge reflections become ignorable.

[Comment 7]: 6. Can authors comment about the possibility of measuring the $W < L$ regime using RuO₂ (or other material)? The resolution of the features proposed (10 nm or less) seems challenging from an experimental fabrication perspective.

[Reply 7]: We thank the reviewer for this valuable comment. In the manuscript, we use L to denote the length of the junction and W to denote the width of the junction. We agree with the reviewer that fabricating devices in the $W < L$ regime can be challenging. However, in order to test our predictions, it is not necessary to fabricate the device deeply in this regime. In fact, comparing the results with $W = 2L$ and $W = 10L$ should be enough to see the distinction (i.e., suppression of supercurrent density) for junctions where Cooper pairs moving with large oblique angles dominate the transport. In fact, devices with similar aspect ratios are possible in experiments [see, e.g., Nature 569, 93 (2019) for a planar Josephson junction with $(L, W) = (600\text{nm}, 1\mu\text{m})$].

The length scale of 10 nm we estimated is the *shortest* periodicity for RuO₂ thin films. Note that this is not the size of the junction but the periodicity of I_c oscillations. The periodicity becomes larger in other directions. Moreover, according to our formula, the periodicity depends sensitively on the model parameters, such as the chemical potential and strength of hopping and altermagnetic order. As the altermagnetism has an increasing number of candidate materials (see, e.g., arXiv:2307.10371, arXiv:2307.10369, and arXiv:2307.10364), we believe that there exist candidate materials with significantly longer periodicity. On the other hand, to clearly observe the oscillations of the Josephson supercurrent, the size of the junction should be longer than the oscillation periodicity. Namely, the junction size could be in the order of 100 nm or larger, which is also experimentally feasible [see e.g., Nature Commun. 9, 3478 (2018) for junction devices of $(L, W) = (90\text{nm}, 920\text{nm})$; and Nature 569, 89 (2019) for devices of $(L, W) = (80\text{nm}, 1.6\mu\text{m})$ and $(L, W) = (40\text{nm}, 5\mu\text{m})$].

Reply to Review #3

[General comments]: The manuscript by Song-Bo Zhang et al. reports on the theoretical study of Cooper pairing in an altermagnetic metal in proximity to conventional s-wave superconductors. Many interesting results are reported, in particular concerning Josephson junctions composed of an altermagnetic metal as barrier and conventional s-wave superconducting electrodes. Specifically, oscillations of the damping parameter and $0-\pi$ transitions occur by varying different parameters of the altermagnet barrier, even in absence of

a net magnetization. Moreover, the transport properties are anisotropic, reflecting in some sense the anisotropic features of the altermagnet barrier.

In my opinion, the topic of the manuscript is timely and quite interesting, the analysis is well supported and the paper is well written. Therefore, I recommend publication of the manuscript in Nature Communications, after a few remarks have been considered.

[Reply]: We thank the reviewer for the concise summary and positive assessment of our work. We are also very grateful to the reviewer for the support in publishing our manuscript. All their suggestions have motivated us to improve the manuscript, with appropriate modifications as detailed in the following point-by-point answers.

[Comment 1]: Specifically, the transport properties strongly depend on the interface orientation between the superconducting electrodes and the altermagnet barrier. It turns out that some crucial unconventional behaviours can be observed if there is control on the interface orientation. For instance, the reduction of the critical current density, when reducing the width of the junction, can be observed when the junction is along the direction where the spin splitting vanishes.

As widely reported in the huge literature on anisotropic HTS superconductors and HTS junctions, control on junction's interface can be quite difficult to achieve.

[Reply 1]: We thank the reviewer for the constructive and insightful comments.

We agree with the reviewer that it might be challenging to rotate the junction orientation in experiments. However, this is not impossible, as we mentioned in the Reply to Reviewer #2. In fact, using the photolithography and ion milling techniques, one can fabricate curved devices with a series of accurate leads that can effectively vary the junction orientation continuously (i.e., by changing the pairs of leads at different positions), see a sketch in Fig. R5 below. Notably, these techniques have been applied successfully to probe the anisotropic properties in magnetic materials, see e.g., Phys. Rev. Lett. **128**, 247202 (2022) and Adv. Electron. Mater. **9**, 2300049 (2023). By replacing the normal metallic leads with conventional superconducting leads such as aluminium and niobium, it is possible to achieve the $0-\pi$ transition by effectively changing the junction orientation.

In the revised manuscript, we have added a sentence “*Experimentally, one could fabricate curved devices with a series of superconducting lead pairs, similar to those used for anisotropic magnetoresistance measurements [41,42], which allows an effective rotation of the junction orientation*” to point out the feasibility of effectively varying the junction orientation.

Fig. R2. Sketch of a curved device. Changing the pairs of leads at different positions, this allows an effective rotation of the junction orientation. Adapted from Adv. Electron. Mater. 9, 2300049 (2023).

[Comment 2]: Moreover, non-uniformity in the barrier along the junction width can induce a mixing of the transport properties, since the interface direction is not the same along the width. For these reasons, some of the experiments proposed by the Authors can be hard to provide clear results.

[Reply 2]: We agree with the reviewer that non-uniformity may occur in materials, which can affect superconducting junctions. However, unlike high-temperature superconductors such as cuprate superconductors, which are susceptible to disorders, our devices involve simply conventional s-wave superconductors, which are generally more robust to disorders. Altermagnetism, on the other hand, arises directly from the symmetries of the crystal potential and does not require strongly correlated systems. Thus, we expect our results based on the long-wavelength model to be less sensitive to non-uniformity in the system, at least at the atomic level.

[Comment 3]: In this sense, phase sensitive experiments can be more powerful, as for HTS junctions. Therefore, it would be very useful if the Authors can provide predictions concerning the behaviour of the mentioned junctions in presence of magnetic field, the magnetic field pattern of the critical current, or the behaviour in a SQUID geometry. This would provide much more powerful insights to be tested in an experiment.

[Reply 2]: We thank the reviewer for this constructive suggestion. Motivated by the reviewers' comments, we have further investigated the Fraunhofer interference pattern in the presence of an external magnetic field. Strikingly, we find that the junctions oriented along the maximum and vanishing spin-splitting directions exhibit distinct Fraunhofer patterns due to their different dominant Cooper-pair transfer trajectories. In particular, at the $0-\pi$ transition point of the junction oriented along the direction where spin-splitting vanishes, the critical supercurrent can be induced and enhanced by an applied magnetic field, as shown in Fig. R6. In contrast, in the junction oriented along the direction of maximal splitting, we observe a Fraunhofer pattern,

in which the maximum critical supercurrent is at zero field. This contrast may provide us with another compelling signature to detect the unique superconducting transport properties of altermagnetic junctions. The phase coherence effect in a SQUID geometry is also a very interesting point. However, this problem is quite involved study in our formalism as it involves too many integrations. We may defer it to future study.

In the revised manuscript, we have provided a new section of the Fraunhofer pattern with the new figure (Fig. 5) and added the relevant calculation to the Supplementary Information (Sec. IV "Calculations of the Fraunhofer pattern").

Fig. R6. (a) Maximal supercurrent I_s as a function of chemical potential μ and an external perpendicular magnetic flux (field) $\Phi = BWL$ (in units of magnetic flux quantum h/e) in a Josephson junction oriented along x - or y -axis. (b) The same as (a) but for a junction oriented with $\varphi = \pi/4$. (c) Fraunhofer patterns at a 0 - π transition point (pink) or away from the transition points (blue) in a junction oriented along x - or y -axis. The corresponding chemical potentials are marked by the colored markers in the inset. The two curves correspond to the two dashed line cuts in (a), respectively. (d) The same as (c) but for a junction oriented with $\varphi = \pi/4$. Other parameters are the same as Fig. 3(a) in the main text.

List of main changes

We marked the main changes of the revised manuscript and Supplementary Information in blue color.

1. We have added the explanation “*In Josephson junctions, the ground state usually has no phase difference across the junction and is referred to a 0-junction. However, the finite magnetization may produce an intrinsic π phase difference, forming a so-called π -junction [17,18]. Notably, a switchable π state of the Josephson junction holds important applications in superconducting circuits and qubits [19-21]*” to the end of the first paragraph.
2. We have added a sentence in the introduction “*It breaks the combined symmetry of translation and C_2 spin rotation that flips the spin direction (which is required for classical collinear antiferromagnets), but preserves a combined symmetry of spatial rotation and C_2 spin rotation*” to distinguish the altermagnetic and anti-ferromagnetic materials.
3. We have added the sentence to the caption of Fig. 1 “*In the calculation for the ferromagnet, we replace the altermagnetic order with a constant magnetization in Eq. (1) and consider the long-wavelength limit with rotation symmetry in the normal kinetic energy part.*”
4. We have rephrased the description of the symmetry of the model as “*The model respects [$C_2||C_{4z}$] symmetry, i.e., a four-fold rotation in real space $(x, y) \rightarrow (y, -x)$ together with a two-fold rotation in spin space $(\uparrow, \downarrow) \rightarrow (\downarrow, \uparrow)$, which is indicated by the relation $s_y \mathcal{H}^*(\mathbf{k}) s_y = \mathcal{H}(k_y, -k_x)$.*”
5. We have added a comment on the side boundary reflections “*Here, we ignore the correction from side boundary reflections. This is justified for $L \lesssim W$, as we show in the Supplementary Information [40]*” and the referred to the new calculation for side boundary reflections to the Supplementary Information (Sec. V “Correction by side boundary reflections”).
6. We have rephrased the sentence in the first paragraph of page 5 “*This suggests that the measurement of I_c gives direct access to the information about the order parameter*” as “*This suggests that the measurement of the supercurrent I_c , as a function of junction length, chemical potential, and junction orientation can give access to the information about the order parameter, such as its dependence on position (measured from the AM-SC interface), chemical potential and junction orientation*”.
7. We have added a sentence “*Experimentally, one could fabricate curved devices with a series of superconducting lead pairs, similar to those used for anisotropic magnetoresistance*”

measurements [41,42], which allows an effective rotation of the junction orientation” and new references numbered by 41 and 42.

8. We have added a description of the calculation “... *our numerical calculations in Figs. 4(b) and 4(d), where we perform the direct integration of x' and x'_1 from $-W/2$ to $W/2$ in Eq. (9)*” at the end of the section of *Dominant Cooper-pair transfer trajectory* in the main text.
9. We have provided a new section and a new figure (Fig. 5) for the results of Fraunhofer interference patterns in the main text (on page 6) and added the relevant calculation to the Supplementary Information (Sec. IV "Calculations of the Fraunhofer pattern").
10. We have corrected the order parameters of RuO₂ and KRu₂O₈ as $k_x k_y s_z$ and $(k_x^2 - k_y^2) s_z$ and referred to Ref. [23] explicitly. We have also mentioned explicitly that these two types of altermagnetic order are related to each other precisely by an axis rotation of 45 degrees.
11. We have added a comment on spin-orbit coupling to the discussion section “*it would be interesting to extend our study to the case with valley degrees of freedom in the altermagnet or involving triplet pairing, in particular, in the presence of spin-orbit coupling.*”
12. We have modified the abstract, the introduction and discussion sections to emphasize the new results about Fraunhofer patterns.

REVIEWERS' COMMENTS

Reviewer #1 (Remarks to the Author):

The authors have answered satisfactorily our questions. We also appreciate the extra Fraunhofer pattern calculations shown in new Fig. 5 which are further highlighting the interesting behavior of Josephson junctions with altermagnets. We only suggest adding missing relevant references: Among citing references [22-24] should be referenced also another two relevant papers which appeared public before the ref. [22-24]: arXiv:1901.00445, Sci. Adv. 6, eaaz8809 (2020) – currently missing – relevant work on RuO₂, arXiv:1902.04436, PRB 99, 184432 (2019) – currently cited only as Ref. 28. Providing the references are corrected we have no other suggestion, are recommending the work for publication and believe it will represent important contribution to the new field.

Reviewer #2 (Remarks to the Author):

Authors have provided a detailed reply to all my questions, including modifications of the main text. I specially appreciate the new results related to the Fraunhofer pattern. In my opinion, these features provide a more robust evidence of the finite momentum of Cooper pairs than the previously proposed 0- π transition. Also, I appreciate their explanation related to the difference to antiferromagnets. Overall, I think the work has been improved significantly since last version, fitting, in my opinion, the publication criteria of Nat. Commun. For these reasons, I would like to recommend the article for publication.

Reviewer #3 (Remarks to the Author):

In the revised version of the manuscript, the Authors have convincingly answered to all my previous criticisms, as well as the concerns by the other referees. In particular, the study of the magnetic field patterns provides clear fingerprints for future experiments and is of fundamental value for further studies in the field.

In my opinion, considering the impact of the work and the scientific advance with respect to the existing literature, the manuscript deserves to be published in Nature Communications.

Response to the Reviewers

Nature Communications NCOMMS-23-33963A

Reply to Review #1

[Comments]: The authors have answered satisfactorily our questions. We also appreciate the extra Fraunhofer pattern calculations shown in new Fig. 5 which are further highlighting the interesting behavior of Josephson junctions with altermagnets. We only suggest adding missing relevant references: Among citing references [22-24] should be referenced also another two relevant papers which appeared public before the ref. [22-24]: arXiv:1901.00445, Sci. Adv. 6, eaaz8809 (2020) – currently missing – relevant work on RuO₂, arXiv:1902.04436, PRB 99, 184432 (2019) – currently cited only as Ref. 28. Providing the references are corrected we have no other suggestion, are recommending the work for publication and believe it will represent important contribution to the new field.

[Reply 1]: We thank Reviewer #1 for pointing out the lack of the relevant references and recommending our manuscript for publication. In the revised manuscript, we have added all the references as you suggested.

Reply to Review #2

[General comments]: Authors have provided a detailed reply to all my questions, including modifications of the main text. I specially appreciate the new results related to the Fraunhofer pattern. In my opinion, these features provide a more robust evidence of the finite momentum of Cooper pairs than the previously proposed 0- π transition. Also, I appreciate their explanation related to the difference to antiferromagnets. Overall, I think the work has been improved significantly since last version, fitting, in my opinion, the publication criteria of Nat. Commun. For these reasons, I would like to recommend the article for publication.

[Reply]: We thank Reviewer #2 for his/her positive comments and recommendation for publication in Nature Communications.

Reply to Review #3

[General comments]: In the revised version of the manuscript, the Authors have convincingly

answered to all my previous criticisms, as well as the concerns by the other referees. In particular, the study of the magnetic field patterns provides clear fingerprints for future experiments and is of fundamental value for further studies in the field.

In my opinion, considering the impact of the work and the scientific advance with respect to the existing literature, the manuscript deserves to be published in Nature Communications.

[Reply]: We thank Reviewer #3 for his/her positive comments and recommendation for publication in Nature Communications.